# First Report on Microcystin-LR Occurrence in Water Reservoirs of Eastern Cuba, and Environmental Trigger Factors

**DOI:** 10.3390/toxins14030209

**Published:** 2022-03-15

**Authors:** José Carlos Rodríguez Tito, Liliana Maria Gomez Luna, Wim Noppe Noppe, Inaudis Alvarez Hubert

**Affiliations:** 1Faculty of Chemical Engineering and Agronomy, University of Oriente, Santiago de Cuba 90400, Cuba; lilianag@uo.edu.cu; 2National Centre of Applied Electromagnetism, University of Oriente, Santiago de Cuba 90400, Cuba; 3Department of Biology, Aquatic Biology, KU Leuven Kulak, E. Sabbelaan 53, 8500 Kortrijk, Belgium; wim.noppe@kuleuven.be; 4Laboratorio de Microbiología Molecular y Biotecnología Ambiental, Departamento de Química, Universidad Técnica Federico Santamaria, Valparaiso 2390123, Chile; inaudis.alvarez@sansano.usm.cl

**Keywords:** cyanobacteria, TN:TP, microcystin-LR, UPLC–MS, water reservoir

## Abstract

The factors related to cyanotoxin occurrence and its social impact, with comprehension and risk perception being the most important issues, are not yet completely understood in the Cuban context. The objectives of this research were to determine the risk extension and microcystin-LR levels, and to identify the environmental factors that trigger the toxic cyanobacteria growth and microcystin-LR occurrence in 24 water reservoirs in eastern Cuba. Samplings were performed in the early morning hours, with in situ determination and physicochemical analysis carried out in the laboratory. Microcystin-LR were determined in water and within the cells (intracellular toxins) using UPLC–MS analysis after solid phase extraction. The reservoirs studied were found to be affected by eutrophication, with high levels of TN:TP ratio and phytoplankton cell concentrations, high water temperatures and low transparency, which cause collateral effect such as cyanobacterial bloom and microcystin-LR occurrence. In Hatillo, Chalóns, Parada, Mícara, Baraguá, Cautillo, La Yaya, Guisa and Jaibo reservoirs, concentrations of MC-LR higher than the WHO limits for drinking water (1 µg·L^−1^), were detected.

## 1. Introduction

The increasing eutrophication of aquatic environments caused by industrial development favors the massive growth of microorganisms such as cyanobacteria [1]; these species are able to produce potent toxins (i.e., microcystins and cylindrospermopsins) [2], which affect drinking water and aquatic organisms through the food chain [3,4]. The consequences of the poisoning or damage caused by cyanotoxins are often being addressed in the medical world, leading to large economic costs [5]. Some environmental factors such as pH, spectral irradiance and temperature can influence the proliferation of toxic phytoplankton species in water bodies [6,7]. Additionally, eutrophication occurs due to agriculture and urbanization, whereby some cyanobacteria species can multiply rapidly and form blooms [8]. 

Moreover, the current changing climate could potentially favor toxic cyanobacteria growth [6,9,10]. The frequency of cyanotoxin occurrence as well as the lack of strategies to prevent and/or alleviate cyanotoxin has led to severe impacts on animal and human health, which are a global concern [11,12]. Microcystin is the most studied cyanotoxin with more than 90 variants, microcystin-LR (MC-LR) being the most toxic and widely-distributed [13].

It is crucial to note that water coverage is one of the most important indicators for human development [14]. Water reservoirs, due to their social impact, are a relevant issue in terms of cyanotoxin risk management; their systematic monitoring plan is necessary, as well as policy actions to prevent damage to human and/or animal health. 

In Cuba, even though irrigation is identified as the main use for water reservoirs, the drinking water supply has the most important and complex use because it needs water treatment availability and management in order to comply with safe water quality standards [15]. Water issues are very important at national level due to the prognosis of droughts in the immediate future [16] and the trophic status of the water sources [17].

In Cuban legislation, there are gaps concerning cyanobacteria and cyanotoxin occurrence and its potential impacts. Even though cyanotoxin levels in water reservoirs need to be analyzed, there is some evidence of potentially toxic cyanobacteria occurrence [17,18]. Therefore, there is not any mention of this problem in the established national rules regarding analyzing the water quality in relation to water reservoir uses, as well as there being a low perception concerning this risk [18,19]. Likewise, research dealing with toxic cyanobacteria is limited in Cuba, but there are some highlights from studies carried out in the eastern part of the country [20]. 

Globally, the presence of *Microcystis*, *Oscillatoria* and *Anabaena* toxic species in water reservoirs has frequently been reported [21,22,23,24], with the *Microcystis* dominance being remarkable. Evidence obtained in reservoirs of Santiago de Cuba province describes in the early studies the occurrence of *Microcystis viridis*, *Aphanothece minutissima*, *Oscillatoria chalybea*, *O. limosa*, *O. tenuissima*, *Anabaena torulosa*, *Planktothrix* sp., *Lyngbya* sp. and *Synechococcus* sp., in three water reservoirs (Charco Mono, Chalóns and Parada). The occurrence of some potentially toxic species were afterwards described in other reservoirs, and the *Microcystis* genus was the best represented, with six species: *M. aeruginosa*, *M. comperei*, *M. flos-aquae*, *M. panniformis*, *M. viridis* and *M. wesenbergii* [25]. In order to understand the extension of the risk, other reservoirs needed to be studied, and the main recommendation of this previous research was to determine the level of main toxins associated with the previously identified toxic cyanobacteria.

Moreover, the factors related to cyanotoxin occurrence and its social impact are not yet completely understood [17,18] since some knowledge gaps remain in the Cuban context; some of the most important are those previously mentioned (the comprehension of the risk extension and main cyanotoxin levels). The objective of this paper is to identify the risk extension and the main environmental factors that trigger the toxic cyanobacteria growth and microcystin-LR levels in selected water reservoirs of eastern Cuba. 

## 2. Results

### 2.1. Catchment Description

The main activities in the studied catchment areas were forestry (68%) and agriculture (30%). Other activities such as industry and mining are less important (2%). The highest agriculture use in the catchment area was identified in Bio (90%) and La Yaya reservoirs (70%), while the highest forestry use (80%) was identified in Paso Malo and Charco Mono. Deforested areas were present in all the studied reservoirs, Chalóns reservoir being the highest (30%), while Cautillo, Hatillo and Bio reservoirs showed deforestation of only 5% (Appendix A).

### 2.2. Physicochemical Parameters

The pH values registered in all the study reservoirs were slightly alkaline, ranging from 7.49 to 8.45 units. The conductivity average for the reservoirs was 685 µS·cm^−1^. The lowest conductivity was measured in Gota Blanca (445 µS·cm^−1^), while the highest value was observed in Parada reservoir (1174 µS·cm^−1^) (Figure 1). The dissolved oxygen (DO) was measured in the surface water (50 cm of the water column). It has an average concentration of 6.9 mg·L^−1^, and the lowest values were measured in Parada reservoir, with 3.7 mg·L^−1^. 

The average transparency for all the water reservoirs was 0.9 m. The minimum transparency was measured in Camazán reservoir (0.5 m), and the maximum transparency was measured in Gota Blanca (1.8 m) (Figure 1). The chlorophyll *a* concentration average was 53.5 µg·L^−1^, the maximum values were obtained in Chalóns (127.4 µg·L^−1^), and the lowest concentration was in Mícara (19.3 µg·L^−1^) (Figure 1).

Concerning nutrient concentration, the total nitrogen average (TN) in the studied reservoirs was 1.0 mg·L^−1^, with a minimum concentration in Gilbert (0.3 mg·L^−1^) and a maximum in Mícara (2.5 mg·L^−1^). The TN values were higher than 1.5 mg·L^−1^ in Parada, La Yaya, Cautillo and Gibara reservoirs (Figure 2). The average of the total phosphorus concentration (TP) for all reservoirs was 0.1 mg·L^−1^, and Los Plátanos was the reservoir with the lowest concentration (0.01 mg·L^−1^), while Cautillo reservoir has the highest values (0.2 mg·L^−1^) (Figure 2). In regard to the ratio TN:TP, the highest atomic ratio was obtained in Los Plátanos reservoir (TN:TP = 68), and the lowest in Chalóns (TN:TP = 7).

### 2.3. Phytoplankton/Cyanobacteria

Phytoplankton cell concentration average was 8.4 × 10^4^ cell·mL^−1^. The highest phytoplankton counting was recorded in Parada reservoir (26 × 10^4^ cell·mL^−1^), and the lowest in Gota Blanca (2.0 × 10^4^ cell·mL^−1^). Specifically, the cyanobacteria average concentration was 3.4 × 10^4^ cells·mL^−1^ with maximum densities in Mícara (19 × 10^4^ cell·mL^−1^), and the lowest in Moa reservoir (0.28 × 10^4^ cells·mL^−1^). Chalóns (9.8 × 10^4^ cells·mL^−1^), Hatillo (9.4 × 10^4^ cells·mL^−1^) and Baraguá (8.6 × 10^4^ cells·mL^−1^) reservoirs had concentrations of cyanobacteria cells near the 10^5^ cells·mL^−1^ (Figure 3).

In Mícara reservoir, the highlight is that the entire composition of phytoplankton community were cyanobacteria, while in Baraguá and Chalóns, the percentages of cyanobacteria in relation to phytoplankton counting were 86 and 82%, respectively. Hatillo revealed a percentage higher that 50%, but the rest of the reservoirs were below 40%. 

During this research a total of 86 phytoplankton species were identified, 38 of which were cyanobacteria (44%). Chlorophyta and Heterokontophyta divisions had 23 and 19% cyanobacteria, respectively. The Bacillariophyta were not well represented; they comprised only 2% of the species. 

Twenty-three species of potentially toxic cyanobacteria were identified, which represents 61% of the total cyanobacteria species. The list of cyanobacteria species and potentially toxigenic species is presented in Appendix A [26]. The highest cyanobacteria abundance was found in Baraguá reservoir with 20 species, 13 of which were potentially toxic. The lowest was registered in Paso Malo reservoir with only three species, two of which were potentially toxic [11,27,28,29,30]. 

The most frequent species in the reservoirs were *Synechocystis aquatilis* and *Aphanocapsa* sp.; both were present in 15 reservoirs (62.5%), followed by *Planktothrix* sp. and *Merismopedia punctata* (54%); while *Romeria simplex* was still present in 50% of the reservoirs studied (Appendix A).

Five species were unique for different reservoirs, i.e., *Chroococcus minutus* only occurred in Joturo, *Coelosphaerium* sp. in Mícara, *Spirulina subsalsa* in Baraguá; *Coelosphaerium kuetzingianum* and *Woronichinia* sp., both occurred only in the Charco Mono reservoir. The genus *Microcystis* had the highest species diversity with six species (Appendix A) [26].

Different blooms were visually detected in 33% of the reservoirs studied, highlighting *Microcystis* species blooms of *M. aeruginosa*, *M. panniformis*, *M. wesenbergii* and *M. flos-aquae bloom. Cylindrospermopsis curvispora*, *C. raciborskii*, *Oscillatoria* sp., *Synechocystis aquatilis*, *Synechococcus* sp. and *Planktolyngbya limnetica* blooms were detected too. The main bloom-forming species in the studied reservoirs played a role in toxic events affecting animals and humans worldwide [26,31,32,33,34].

### 2.4. Microcystin-LR Concentration

The methodological accuracy to the MC-LR quantification was assessed through a calibration curve carried out in UPLC–MS, with a linear relationship in the calibration concentration range. Highest values of microcystin-LR (Table 1) were detected in biomass samples (intracellular), specifically in Cautillo (65.8 µg·L^−1^), La Yaya (60.5 µg·L^−1^) and Mícara reservoirs (55.8 µg·L^−1^). In general, water samples exceeded the World Health Organization (WHO) MC-LR limits for drinking water supply (1 µg·L^−1^) in nine reservoirs (29%), Cautillo and La Yaya reservoirs being significantly higher (*p* < 0.05). 

### 2.5. Correlation Analysis

The analysis revealed a negative correlation between air temperature and DO (r = −0.51) and transparency (r = −0.41) (Table 2). Furthermore, a positive correlation was found between phytoplankton cell concentration and air temperature (r = 0.47). A slightly positive correlation between transparency and DO (r = 0.41) was detected too (Table 2), suggesting that oxygen consumption occurred by phytoplankton or chemicals, reducing nutrients in the water [35].

In addition, Secchi disk had a negative correlation with water temperature (r = −0.42) and with chlorophyll *a* (r = −0.41), corroborating the usefulness of these parameters to evaluate water quality and ecosystem trophic status [36,37], which supported the TSI calculation and its analyses [17].

Additionally, phytoplankton cell concentration correlated positively with TP (r = 0.42) and cyanobacteria cell concentration (r = 0.53). Finally, conductivity has negative correlation with cyanobacteria cell concentration (r = −4.46) and chlorophyll *a* (r = −0.45) (Table 2). According to neural network analysis, the variables phytoplankton, TN:TP ratio and water temperature have 60% of normalized importance, which has a high impact on the development of subsequent studies analyzing their direct influence on MC-LR concentration in water and intracellularly.

## 3. Discussion

There are several reports on environmental factor influences regarding the occurrence of cyanobacteria and cyanotoxins around the world. Some of these studies were conducted in the tropical and subtropical regions of Asia [38,39,40,41], However, the studies describing the island conditions in the Caribbean Sea, even in Latin America, are still insufficient [17,20,25,42]. Studies using a combination of correlation, principal component analysis (PCA) and chemical analysis have been presented to confirm these relations [7,21,43,44,45,46,47,48].

The eastern part of Cuba has environmental conditions which might favor the cyanobacteria proliferation, including the nutrient enrichment of aquatic systems, warm temperatures and calm stable water conditions. In all the reservoirs studied, an aquaculture was developed, with an important contribution to the nutrient enrichment [19], water eutrophication being the main problem, related to cyanobacteria and cyanotoxin production [17].

PCA analysis revealed that component 1 is strongly influenced by water and air temperature, TP, phytoplankton and cyanobacteria concentration cells. Component 2 is positively related with TN and TN:TP ratio, and negatively with pigments (chlorophyll *a* and phycocyanin (PC). The first axis shows 32.77% of total variance and the second axis shows 20.92%, with 53.69% of the total variance explained.

Two groups of reservoirs were highlighted: one is the most influenced by component 1, which included 10 water reservoirs with MC-LR concentrations higher than the WHO limits, except for Charco Mono and Camazán reservoirs (Figure 4). Mícara, Parada, La Yaya and Cautillo were positively influenced by both principal components. The other group of 14 water reservoirs, where MC-LR levels were very low or not detected, showed no influence of component 1 (Figure 4). In this group, Jaibo reservoir is included with an intracellular MC-LR concentration of 1.41 µg·L^−1^.

PCA indicates that phytoplankton and cyanobacteria cell concentrations combined with TP and temperature could trigger high concentrations of MC-LR.

Parada, Baraguá, Camazán and Cautillo were the reservoirs with the highest water temperature, but Camazán did not show any cyanobacteria bloom, neither MC-LR concentrations were higher than the drinking water OMS limit. Parada reservoir, coincidently, had the highest phytoplankton concentration, and one of the lowest cyanobacteria diversities, which could be related to bloom conditions. Similar behavior was identified in La Yaya, Guisa and Hatillo reservoirs. These results strengthen the fact that water temperature is commonly related with cyanobacteria proliferation, attaining maximum temperatures above 25 °C [6] and suggesting that elevated temperatures yield more toxic *Microcystis* sp. cells, as indicated in the research by [49], and the incoming scenario with climate change potentially yielding more toxic blooms [9,50,51].

Even though it is not yet clear how phytoplankton will respond globally to the increase in temperature, it is evident that a possible consequence is the reduction of diversity, favoring the growth of different cyanobacteria species, for instance, *Planktothrix* sp. and *Microcystis* sp., which have better survival strategies in new climatic conditions [52,53,54].

Furthermore, it has been demonstrated that in situ toxin release from harmful cyanobacteria increased under elevated temperatures (between 20 and 25 °C) [55]. During this study, even the minimum water temperature value measured (23 °C) is in the optimal range for cyanobacteria growth and toxin release. The major MC-LR concentration in water was recorded in Cautillo, La Yaya, Mícara, Hatillo and Baraguá reservoirs, which show high water temperatures (24.9–31.0 °C).

Parada reservoir presented the lowest DO concentration, but other reservoirs have DO values within the range 5 to 6 mg·L^−1^ (Chalóns, Baraguá and Cautillo), all of them containing cyanobacteria blooms. The former confirms the results of previous reports, which explained that DO diminishes during cyanobacteria blooms [35].

Consequently, the deteriorating quality of water in reservoirs accelerates the ecosystem damage and water functions, which endangers the use of water resources [35,56], producing a negative effect on ecosystem services, mainly habitat, supply, regulation and substrate services.

The low transparency measured in the reservoirs indicates poor water quality, which reveals that high turbidity is most probably associated with cyanobacteria abundance [7]. Another relevant aspect is the pH values, with averages that were 8.02 ± 0.05. Even when pH values are within the limits recommended by the Cuban and WHO regulations for water quality issues (6.5–8.5) [57,58], some authors expressed these values coincide with the optimal pH range because in pH higher than 8 units, HCO_3_^−^ is the main available carbon source, which favors cyanobacteria growth [59,60].

Concentration of chlorophyll *a* was higher than 100 µg·L^−1^ in Camazán and Chalóns reservoirs, which show the worst conditions concerning the trophic status in recent studies [17]. Although chlorophyll *a* is considered a good indicator of trophic conditions in water reservoirs, positively correlated with poor water quality [61], the present research assumes a good indicator of high concentrations of cyanobacteria is not necessarily chlorophyll *a* [62]. This was corroborated in Mícara and Baraguá reservoirs, which had high concentrations of cyanobacteria and MC-LR, and had the lowest chlorophyll *a* concentration.

All the reservoirs studied were found to be eutrophic or hypertrophic when the TP or TN concentrations were analyzed. Additionally, all the reservoirs had phytoplankton concentrations higher than 1500 cell·mL^−1^, which is considered another indicator of eutrophy [27,28,63]. The highest TP concentrations were measured in Hatillo, Camazán, Baragua, Chalóns, Mícara and Cautillo reservoirs (Figure 2), while TN was higher in Mícara and Parada as compared with the other reservoirs.

Sixteen reservoirs showed TN:TP atomic ratios higher than 20, especially Cautillo, Guisa, Hatillo, Jaibo, La Yaya, Mícara and Parada reservoirs, with MC-LR concentrations at the upper OMS guideline limit value of 1.0 μg·L^−1^, advisable for drinking waters [64]. Though some authors have found the TN:TP ratio not to be significant in bloom-forming cyanobacteria occurrence in most tropical reservoirs with high concentrations of total nutrients [47], the results obtained in this research demonstrated the usefulness of this ratio in following the bloom occurrence and cyanotoxin risk management.

Two reservoirs showed TN:TP ratios lower than 20, and MC-LR concentrations, both intracellular and in water samples higher than 1 µg·L^−1^: Chalóns reservoir showed the lowest TN:TP atomic ratio (7), but its cyanobacteria concentration, mainly due to *Synechocystis aquatilis* bloom, was 9.8 × 10^4^ cyanobacteria cells·mL^−1^; similar behavior was found in Baraguá reservoir (26 × 10^4^ cyanobacteria cells·mL^−1^) with a TN:TP atomic ratio (11), associated with a bloom involving *Cylindrospermopsis curvispora* and *Oscillatoria* sp. This result points to the need for risk management related to cyanobacteria growth and cyanotoxin production, and suggests that low TN:TP should not favor microcystin-producing cyanobacteria [65].

TN:TP atomic ratio analysis is useful as an indicator of cyanobacteria risk, according to different authors, but this analysis needs to consider other variables, such as phytoplankton and cyanobacteria cell concentrations, diversity and bloom occurrence [66].

The highest values of MC-LR in water were detected in Hatillo, Baraguá, Chalóns, Mícara and Cautillo reservoirs, with high TP concentrations; the other two reservoirs, Parada and Mícara, showed high TN values. In those reservoirs less enriched with nitrogen and phosphorus, such as Los Plátanos and Clotilde, MC-LR were not detected in water. This means that TN and TP are important variables to consider into the analysis and management of cyanotoxin risk. Additionally, it is important to note that TP was involved in TSI analyses.

Another important aspect in this research is the variation of the phytoplankton concentration. Reservoirs with phytoplankton concentrations higher than 1500 cell·mL^−1^ were considered eutrophic by different authors [27,28,63]. Results reveal that all the water reservoirs studied presented concentrations which exceeded this value. The lowest phytoplankton concentration (2 × 10^4^ cell·mL^−1^) was measured in Gota Blanca reservoir; concentrations higher than 15 × 10^4^ cell·mL^−1^ were measured in Guisa, La Yaya, Hatillo, Mícara and Parada reservoirs.

Concerning cyanobacteria, maximum cell densities were recorded in Mícara (19 ± 2.1 × 10^4^ cell·mL^−1^); Hatillo, Chalóns and Baraguá had cyanobacteria concentrations higher than 7 × 10^4^ cell·mL^−1^. Regarding cyanobacteria percentage, besides Mícara reservoir (100%), Baraguá and Chalóns reservoirs are highlights, with more than 80% of cyanobacteria. The cyanobacteria percentage was higher than 40% of the total phytoplankton in 70% of the reservoirs. The Mícara reservoir is a special case with a *Microcystis* species blooms, which imply the need to develop more studies in relation to environmental variable fluctuation, the seasonal changes, sediment features, and stratification profiles.

The main bloom-forming species identified during this study were *Cylindrospermopsis raciborskii*, *Microcystis aeruginosa*, *Microcystis flos-aquae*, *Microcystis panniformis*, *Microcystis wesenbergii*, *Oscillatoria* sp., *Planktolyngbya limnetica* and *Synechocystis aquatilis*, which are involved in toxic events, affecting animals and humans worldwide [31,32,33,34]. It has been demonstrated that not all the species of these genera have the potential for producing MC-LR; likewise, inside a productive species, not all strains are toxin producers [67,68].

Some studies demonstrated that increase in temperature yielded the highest growth rates of toxic *Microcystis* cells, suggesting that future eutrophication and climatic warming may additively provoke the growth of toxic populations in the *Microcystis* sp., leading to blooms with higher microcystin content [49,55], which become a fact in the reservoirs studied, that have *Microcystis* sp. blooms, three involving *M. aeruginosa*, two involving *M. flos-aquae* and one with *M. panniformis* and *M. wesenbergii*, respectively.

Unlike in other studies [69,70,71], the different land uses and cyanobacteria blooms revealed no direct correlation in the present research. Nevertheless, there is a clear tendency showing that water reservoirs with the highest microcystin concentrations also had more than 40% of intense land use, such as agriculture, deforestation, industry and urban pressure.

Temperature, conductivity, total phosphorus and transparency were positively associated with microcystin-producing taxa, such as *Microcystis aeruginosa* by Barros in 2019, while N limitation seems to be an important driver for microcystin production by *M. aeruginosa* [44].

Water temperature is widely reported as an influencing factor in the growth and development of cyanobacteria, and the production of cyanotoxins. High temperatures would not be the cause of cyanobacteria blooms, but they would be associated with other phenomena such as thermic stratification and changes in the depth of the mix zone, which can favor the development of cyanobacteria with gas vesicles [21]. Chorophyll *a*, phycocyanin (PC) and TSI_Chl*a*_ and TSI_P_, as well as eutrophic level had between 20 and 40% of normalized importance.

This study shows that the combination of high-water temperature, high TN:TP ratios and low transparency, even within a pH range inside the guideline values (6.5–8.5) [57,58], can predict cyanobacterial growth and therefore the combination of the above factors could be used to predict cyanobacteria blooms in tropical water reservoirs, TSI being the better indicator to foresee the tendency to cyanobacteria and cyanotoxin occurrence.

## 4. Conclusions

The climatic conditions in eastern Cuba favor the occurrence of cyanobacteria and cyanotoxin in the reservoirs studied. These are affected by eutrophication, evidenced by high levels of TN:TP ratios and phytoplankton cell concentrations, high water temperatures and low transparency, which cause collateral effects such as cyanobacterial bloom and microcystin-LR occurrence. The most affected reservoirs were Hatillo, Chalóns, Parada, Mícara, Baraguá, Cautillo, La Yaya, Guisa and Jaibo, with MC-LR concentrations higher than the drinking water WHO limits (1 µg·L^−1^).

## 5. Materials and Methods

### 5.1. Study Area

This study was conducted from September 2015 to April 2017, in the eastern region of Cuba, including reservoirs of Granma, Santiago de Cuba, Holguín and Guantánamo provinces (Longitude W: −74,134°, −77,738°; Latitude N:19.841°, 20.730°) (Figure 5). Twenty-four water reservoirs were selected for inclusion in this study, and the main selection criteria were the relatively easy access, the main uses as water supply and crop irrigation, their location and local relevance according to their beneficiaries (Appendix A).

### 5.2. Sampling

All the reservoirs were sampled randomly in rainy and dry seasons, in order to evaluate the risk extension of the phenomena. A unique sampling was established during the morning in each reservoir during the first stage of this study, in order to analyze the influence of physicochemical and environmental factors, and the occurrence of toxic cyanobacteria growth and microcystin-LR, presenting evidence for the risk characterization, including the extension and microcystin levels during the sampling.

### 5.3. Catchment Analysis and Land Uses

Catchment areas were identified using the geographic information system (QGIS software, version 3.2.2-Bonn), and SAGA software. Elevation data were defined by with the Digital Elevation Model (DEM) provided by the Shuttle Radar Topography Mission (SRTM) with 90-m resolution [72]. Channel network, flow direction and catchment area were calculated and delineated using tools from SAGA Software. Percentage of different land use categories, including forest, agriculture, industry, urbanized and deforest, were estimated and projected on the watershed and compared to the official digital map [73].

The total catchment area of all reservoirs included in this study is 14,182 km^2^, which corresponds to 55% of the total area of eastern Cuba (25,700 km^2^); 19 of the studied reservoirs belong to the three main watersheds of this region [15]: Toa-Mayarí (4), Guantánamo-Guaso (4) and El Cauto (11 reservoirs). The other five reservoirs have their own catchment area (Figure 5).

The Cauto watershed catchment includes three provinces (Holguin, Granma and Santiago de Cuba). This watershed is the biggest of Cuba with an area of 8969 km^2^ and it is one of the most affected by human activities, containing about 37% of all water resources in the entire country [15]. The Toa-Mayarí catchment includes north of Guantánamo and southeast of Holguin province, but Guantánamo-Guaso catchment only includes the Guantánamo province [74].

### 5.4. Physico-Chemical Analysis

Samples were collected in the early morning hours (8:00–9:00 am) at the water surface layer (0.5 m depth) of the reservoirs. Three locations were established in each reservoir, considering at least one point of the national monitoring water quality program, but other points were located near the reservoir tapping, and/or in relation to foam or the occurrence of cyanobacteria blooms [25].

Water temperature (°C), pH, conductivity (µS·cm^−1^) and dissolved oxygen (DO) (mg·L^−1^), were measured in situ with a multi-parameter water quality meter (YSI ProDSS, USA). Transparency (m) was determined using a 20 cm Secchi-disc. To the in situ chlorophyll measurements, an AquafluorTM handheld fluorometer (Turner Designs, San Jose, CA, USA) was used, previously calibrated with chlorophyll *a* standard (Sigma, Steinheim, Germany) (Chl *a*) correction factor was used, considering the resulting calibration curve equation, which was prepared from serial dilutions of a standard Chl *a* solution in methanol 90% (10 to 1000 μg·L^−1^) [17].

The Cuban Norms 1021:2014 and 25:1999 established to assess the water quality of reservoirs for drinking water supply and fishing interest [58,75] were used in order to evaluate reservoir conditions. 

Total nitrogen (TN) and total phosphorus (TP) were determined after persulfate oxidation to nitrate [76] and orthophosphate [77]. All measurements were carried out in SKALAR SAN++ Segmented Flow Analyzer (Skalar Analytical B.V., Breda, The Netherlands) using SKALAR protocols (503-010Xw/r(+P14) and 155-008w/r+P14). The TN:TP atomic ratio was calculated using the conversion for TN (0.0714) and TP (0.0323), expressing concentrations in μmolL^−1^ [78].

### 5.5. Phytoplankton Analysis and Taxonomic Identification

The phytoplankton samples collected at the water surface were preserved with Lugol 10% solution [79]. Each sample was conserved in a dark environment and cooled during transportation to the laboratory. Samples for toxin detection were frozen immediately (in duplicate) without preservation.

Cyanobacteria counting was performed using a Neubauer hemocytometer chamber and a binocular microscope Olympus (CX31RBSFA, Japan), and percentage of cyanobacteria cell concentration in relation to the total phytoplankton concentration was calculated. Taxonomical identification was performed for species classification, when it was possible, following the expert criteria [80,81,82,83,84] and Algae base support [85].

### 5.6. Microcystin-LR (MC-LR) Analysis

Microcystin-LR determination was performed for both intracellular and extracellular level, using water and filtered samples. Water samples (100 mL) were filtered through Whatman^®^ glass microfiber filters Grade GF/F 25 mm, then, the filters with biomass were frozen at −20 °C [86]. Consequently, the biomass on the filters was suspended in 10 mL of methanol 90% and sonicated with a Branson sonicator 450 (Branson Ultrasonics Corp., Danbury, CT, USA) for 10 min. This process was repeated three times to obtain a sample volume of 30 mL [86]. This solution was dried under a nitrogen stream and re-dissolved in water.

For MC-LR purification, the extracts were centrifuged (Sigma 6-16 KS, Germany) and the supernatant was purified through an Empore™ SPE Cartridge C18, which was previously conditioned with methanol (ROTISOLV^®^ HPLC grade): water mix (3:6). The analyte-adsorbed column was washed sequentially with 6 mL water and 10 mL of MilliQ water: methanol 20% (*v*/*v*), dried at low-pressure. Microcystins then were eluted with 10 mL of MilliQ water: methanol 80% (*v*/*v*) [87]. The elution was dried under a nitrogen stream. The residue was re-dissolved in 1 mL to be analyzed in Ultra High-Pressure Liquid Chromatography–Mass Spectrometry (UPLC–MS) [88,89,90].

The MC-LR was also extracted from filtered water samples (100 mL) through a purification process as described above.

Working standards for the calibration curve to quantified MC-LR were prepared using a stock solution standard from Enzo Life Sciences (5 mg·L^−1^; ≥95%, HPLC). Serial dilutions were applied to produce standard solutions with concentrations ranging from 1 μg·L^−1^ to 100 μg·L^−1^. The calibration curve was based on the peak area in the chromatogram, measured with the software Empower 3.0 against the standard concentrations of microcystin-LR.

Chromatograms for MC-LR detection were performed on a UPLC–MS separation module (Waters, Zellik, Belgium). The separation was achieved via an ACQUITY BEH-C18 column (150 mm length, 2.1 mm internal diameter, 1.7 µm particle size) and BEH VanGuard Pre-column (5 mm length, 2.1 mm internal diameter, 1.7 µm particle size) from Waters (Zellik, Belgium). Methanol (HPLC grade; Acros Organics, Geel, Belgium), acetonitrile (HPLC grade; Biosolve Chemie, Dieuze, France) and formic acid (Merck, Darmstad, Germany) were used. Milli-Q water was obtained using a Millipore purification system.

Columns were kept at 35 °C during analysis. The injection volume was 20 µL. The mobile phase A was Milli-Q water + 0.1% formic acid (LCMS/MS grade; Merck, Germany) and mobile phase B was acetonitrile + 0.1% formic acid (LCMS/MS grade; Merck, Germany). The gradient elution was performed as a variation [91] with a flow rate of 0.4 mL min^−1^ initially with 80% (A) and 20% (B) until minute 3, changing the proportion from minute 3 to 8 to 35% (A) and 65% (B) and keeping it until minute 10. Finally, it was changed to 80% (A) and 20% (B) until minute 13 [91].

Detection and quantification of microcystins were performed in ACQUITY QDa Performance MS Detector (Waters Corporation, Milford, CT, USA) equipped with an electrospray ionization source (ESI), capillary voltage of 0.8 kV, using selected ion recording (SIR) with mass 995.60 Da for MC-LR determination. The source off-set was 15 V and the temperature was 400 °C [88,90].

Validation of the method consisted of defining the linearity of the limit of detection (LOD) and limit of quantification (LOQ). Linearity was assessed by the relationship between the signal (peak area) and the analyte concentration expressed by linear regression of 7 different standard concentrations (Figure 6).

The determination of LOD and LOQ was based on the standard deviation of the response of blank samples injected, and the slope (m) of the calibration curve. LOD was calculated as 3.3 times the standard deviation (S) of the signal-to-noise ratio ($\mathrm{LOD}=3.3\frac{S}{m}$) and LOQ as 10 times the standard deviation (S) of the signal-to-noise ratio $\left(\mathrm{LOQ}=10\frac{S}{m}\right)$, both divided by the slope of the calibration curve (m) [92,93,94,95].

Microcystin-LR recoveries were obtained by spiking 10 μg·L^−1^ of microcystin-LR (Enzo Life Sciences) to water samples (100 mL), and to filters containing phytoplankton biomass. The same methodology was used in both cases. The SPE purification and analysis using UPLC–MS were performed in triplicate.

### 5.7. Statistical Analysis

Mean, variance (ANOVA, Tukey-test), standard deviation, relative error and correlation tests between variables (Pearson, *p* < 0.05), and principal component analysis (PCA), were conducted using R studio (Version 1.2.5019). The data were normalized using Ln (water temperature), Log_10_ (chlorophyll *a*, cyanobacteria and phytoplankton concentration), Ln (1/X) in microcystin-LR (intracellular and extracellular), inverse (conductivity) and square root of arcsine (transparency) transformation. Graphs were produced with SigmaPlot Version (11.0.0.77) software.

A study of neural networks was conducted using IBM SPSS Version 19.0 and Sigmaplot Version (11.0.0.77) software to analyze the normalized importance of the factors that condition the occurrence of microcystins, using 60% training, 30% test and 10% reserve. Standardized methods were used considering 2 factors (season and trophic state index), 7 covariates (phytoplankton, water temperature, TN:TP ratio, chlorophyll *a*, phycocyanin, TSI_Chl*a*_ and TSI_P_ and as target variables, MC-LR) were considered in water and in biomass.

## Figures and Tables

**Figure 1 toxins-14-00209-f001:**
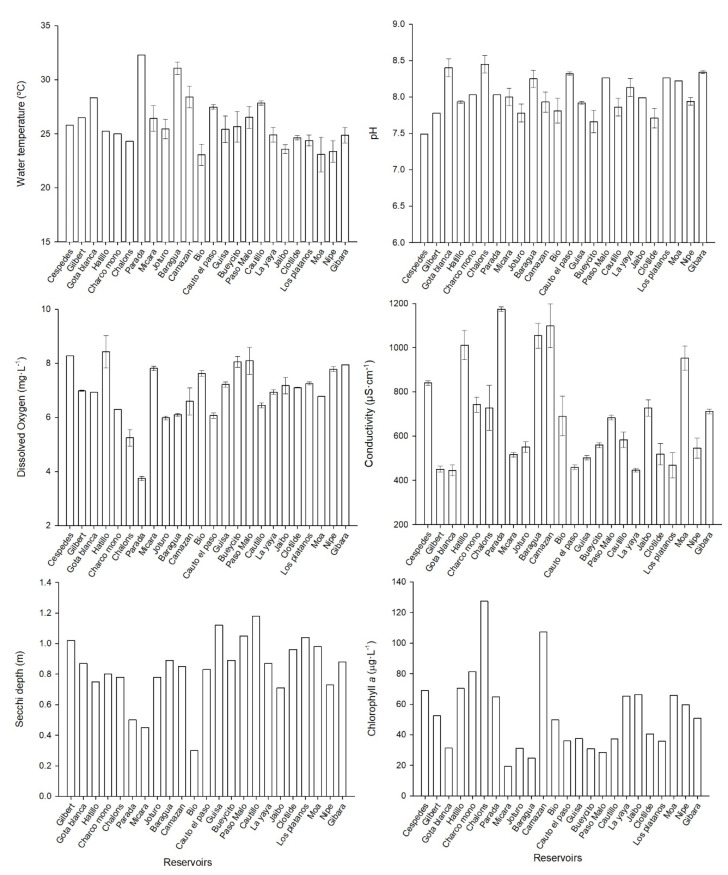
Physicochemical analysis in the studied reservoirs in eastern Cuba.

**Figure 2 toxins-14-00209-f002:**
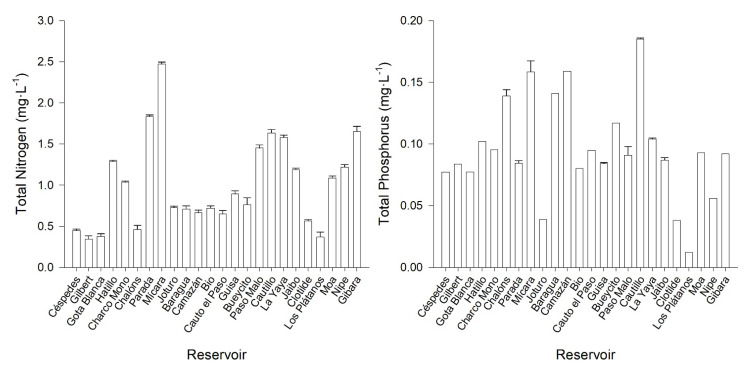
Total nutrient concentrations in the studied reservoirs in eastern Cuba.

**Figure 3 toxins-14-00209-f003:**
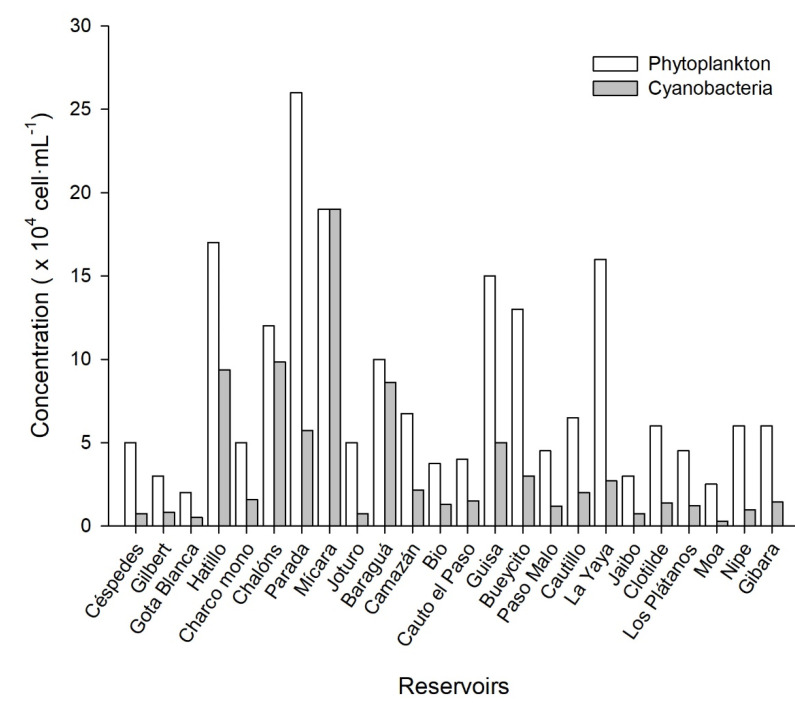
Cyanobacteria cell concentration in 24 studied reservoirs at the eastern of Cuba.

**Figure 4 toxins-14-00209-f004:**
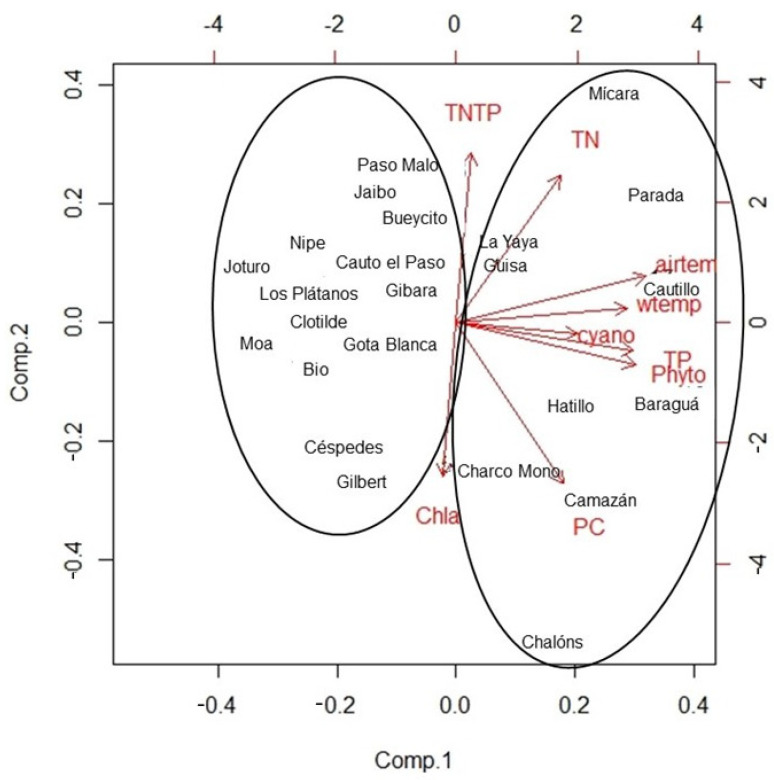
Principal component analysis (PCA) of different environmental variable measures in 24 reservoirs in eastern Cuba.

**Figure 5 toxins-14-00209-f005:**
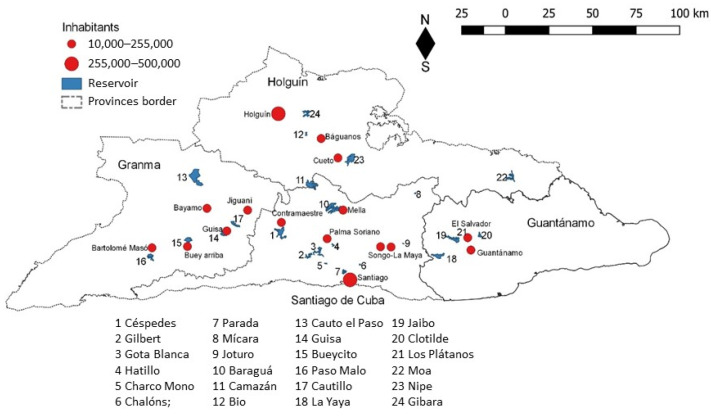
Localization of the selected reservoirs for the study, and associated watersheds.

**Figure 6 toxins-14-00209-f006:**
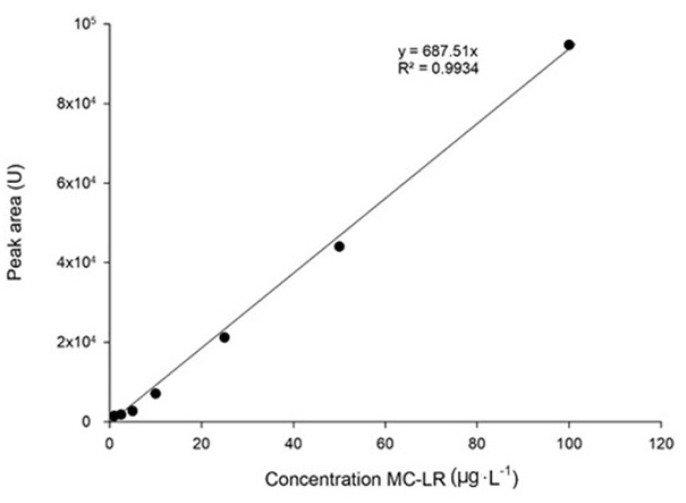
Calibration curve for microcystin-LR (MC-LR) analysis with UPLC–MS.

**Table 1 toxins-14-00209-t001:** Microcystin-LR concentration in studied water reservoirs in eastern Cuba; ND: not detected; bold = concentration higher than 1 µg·L^−1^.

Reservoirs	MC-LR (Extracellular)	MC-LR (Intracellular)
Céspedes	0.04	ND
Gilbert	0.10	ND
Gota Blanca	0.19	ND
Hatillo	**2.38**	**36.50**
Charco Mono	ND	0.015
Chalóns	**2.56**	**10.69**
Parada	**2.21**	**1.93**
Mícara	**5.05**	**55.78**
Joturo	0.35	ND
Baraguá	**1.63**	**17.09**
Camazán	0.14	ND
Bio	0.20	0.06
Cauto el Paso	ND	ND
Guisa	0.95	**1.35**
Bueycito	0.53	0.21
Paso Malo	ND	ND
Cautillo	**14.49**	**65.76**
La Yaya	**14.58**	**60.52**
Jaibo	0.83	**1.41**
Clotilde	ND	ND
Los Plátanos	ND	ND
Moa	ND	ND
Nipe	ND	ND
Gibara	ND	ND

**Table 2 toxins-14-00209-t002:** Correlation of biotic and abiotic parameters in the 24 water reservoirs. (* = significant at *p* < 0.05; ** = significant at *p* < 0.01; *** = significant at *p* < 0.001). Water Temp (Temperature); Air Temp (Air Temperature); Cond (Conductivity); DO (Dissolved Oxygen; Chl *a* (Chlorophyll *a*); Secchi (Transparency); Phyto (Phytoplankton concentration cells); Cyano (Cyanobacteria concentration cells); Phyco (Phycocyanin); TN (Total nitrogen); TP (Total phosphorus); MC-LR Water (MC-LR concentration in water); MC-LR Intrac (intracellular MC-LR concentration).

	Air Temp	Cond	DO	pH	Chl *a*	Secchi	Phyto	Cyano	Phyco	TN	TP	MC-LR Water	MC-LR Intrac
Water temp	0.77 ***	−0.21	−0.51 *	0.12	−0.26	−0.41 *	0.37 *	0.09	0.24	0.11	0.38 *	0.08	0.30
Air temp	-	0.08	−0.37 *	0.01	−0.20	−0.25	0.47 *	0.06	0.22	0.26	0.50 *	−0.11	−0.28
Cond		-	0.19	0.08	−0.45 *	0.36 *	−0.29	−0.46 *	−0.31	−0.10	−0.31	0.14	−0.03
DO			-	−0.29	−0.23	0.41 *	−0.26	0.09	−0.36 *	0.03	−0.12	0.01	−0.26
pH				-	0.04	0.01	0.11	0.26	−0.06	0.06	0.09	−0.60 *	−0.38
Chl *a*					-	0.41 *	0.12	0.01	0.36 *	−0.17	0.06	0.26	0.38
Secchi						-	−0.27	−0.15	−0.12	0.01	−0.39	−0.13	0.01
Phyto							-	0.53 **	0.39 *	0.16	0.42 *	−0.43 *	−0.40
Cyano								-	0.16	0.39 *	0.28	−0.62 **	−0.55 *
Phyto									-	−0.19	0.37 *	0.25	0.14
TN										-	0.39 *	−0.67 **	−0.37
TP											-	−0.18	−0.26
MC-LR water												-	0.91 ***

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
