# Peer review of "First Report on Microcystin-LR Occurrence in Water Reservoirs of Eastern Cuba, and Environmental Trigger Factors"

_toxins, 2022, doi:10.3390/toxins14030209_

Round 1

Reviewer 1 Report

My answer to Authors is in the attached file.

Author Response

Dear Reviewer:

Thank you for your considerations about the paper.

We fixed all suggestions you made related table S1(reference in line 82) and non-toxigenic species in table S2. In line 348 we referenced table S1, but it was with supplement document 1, it was a mistake, now is table S1 in this line.

We had problems to upload table S2 in different document than table S1. Both tables were completed with correct word in columns and rows, as you said.

About high concentration of  MC-LR in Cautillo and La Yaya, could be associated with more toxic strains of Microcystis aeruginosa in both reservoirs and Oscillatoria sp. in Cautillo.

I would like to excuse with you about the mistakes typing text in line 410 about extraction of toxins.

I think now is clear extraction with 90 % methanol, then dried and re-dissolved in water

We are open to receive some other suggestions in order to fix the paper to publish in Toxins.

Thank you

Reviewer 2 Report

The study title is focused on the Microcystin-LR occurrence in water reservoirs of eastern Cuba and environmental trigger factors. This is a very interesting topic to search possibilities to solve problems with cyanotoxins poisoning. There is data that should be improved in this study.

  1. Please use advanced translator and cure this text. There are errors that start from abstract: line 5 under-stood, line 6 and 7 microcystin-LR or Microcystin-LR and others in the text.
  2. Please explain why you used total nitrogen when chemical forms of nitrogen and trace elements play important roles in the production of MC toxins.
  3. Please consider to add why MC-LR was the main area of interest not MC-RR
  4. Please add all limitations of the study.
  5.  Please rewrite references according to MDPI rules.
  6. Please add author contributions.
  7. Please add funding.

Author Response

Dear Reviewer:

Thank you for your considerations about the paper.

We made a deep revision of English, some typing errors were fixed in the text. The reference were rewrite according MDPI rules, using endnote software.

We studied MC-LR study and not MC-RR, because LR is the most toxic and widely distributed.

The data of other forms of nitrogen was not possible because of problems with some samples conservation and results. Fortunately, total nitrogen and phosphorus data, let us to approach to eutrophication process and its associations with cyanobacteria proliferation and toxin production.

In the first version uploaded we add author contributions, we will upload again. Funding we are not able to add, sorry about that.

We are open to receive some other suggestions in order to fix the paper to publish in Toxins.

Thank you

Reviewer 3 Report

 The manuscript describes the presence of cyanobacteria and cyanotoxin in water reservoirs in eastern Cuba. It is well written and structured, scientific relevance is adequate for the journal, so this reviewer recommends minor changes before publication.

Line 29. Please, rephrase

Line 46-49.  Please, rephrase

Figures 1 and 2. Some of the errors are not shown in some graphs. Please review it. If they are so small that they cannot be seen at this scale, you can highlight it in the text or in the figure caption.

Author Response

Dear Reviewer:
Thank you for your considerations about the paper.

As you suggested we made some modifications with English, and rephrase some lines.
Related with errors, in different graphs is cero, for instance with Secchi disk measures. If you consider, we can change all the graphs, and placed one by one in the text with bigger scale, but in some cases error will not be enough big to see.
We are open to receive some other suggestions in order to fix the paper to publish in Toxins.

Thank you

Round 2

Reviewer 1 Report

Dear Authors!

The last suggestions:
-    Use italics for species names (sometimes it is not used)
-    Lines 488-489. Correct the information about Supplementary Materials.

Good luck

Reviewer 2 Report

Accept in present form

This manuscript is a resubmission of an earlier submission. The following is a list of the peer review reports and author responses from that submission.

Round 1

Reviewer 1 Report

Dear Authors!

The goal of this research was to determine the risk extension and the microcystin-LR levels, identifying the environmental factors that trigger the toxic cyanobacteria growth and Microcystin-LR occurrence in 24 water reservoirs at the eastern Cuba.

The good point of the research is an amount of sampling reservoirs. The weak point is discussion. Environmental factors that trigger the toxic cyanobacteria growth remained unclear.

Below I wrote my comment and suggestions:

Section 2.3 These are low abundance values for blooming water bodies. All of them except one are below the level 20,000 cells/ml, which is corresponded to Low Probability of Acute Health Effects according to established WHO guidelines [WHO. 2003. Guidelines for safe recreational water environments. Vol. 1. Coastal and fresh waters. Geneva: World Health Organization. http://www.who.int/water_sanitation_health/ bathing/srwe1/en] It is strange that high amount of microcystins (besides only one structural variant MIC-LR was measured) could be detected.

Figure 5 should be in a “Method” section

Line 168. The Guidelines values proposed by WHO depends on the waterbody type. In a case of recreational use another value should be discussed. The value 1 mkg/L is established for drinking water and could be discussed in a case of drinking water reservoirs and tap water. As I understand, the studied water bodies are nor drinking water reservoir. You could find the suitable values in [WHO. 2003. Guidelines for safe recreational water environments. Vol. 1. Coastal and fresh waters. Geneva: World Health Organization. http://www.who.int/water_sanitation_health/ bathing/srwe1/en]

Lines 153-154 Italics is missed for the species names.

Section 2.5. Some strange correlations are reported (for instance, between temperature and total phosphorus, etc.), it would be better to exclude them. They cannot give any information for the analysis of the obtained data and evaluation of relations.

Figure 6. I would suggest to leave in the table only numbers. This kind of table complicates perception

Line 189 -191 These sentences should be in the discussion part.

Line 195 “There are several reports of the environmental factors influences in occurrence of cyanobacteria and cyanotoxins around the world.” A lot of papers published in the world. It could be a lack of them describing your region?

Line 203-204 Could you describe factors which influence on separation the waterbodies between Component 1 and 2?

Line 225 Temperature optimum is depended on the species type.

Starting from line 267 check the use of the italics for the species names.

Line 277 Usually trophic status is estimated using Chlorophyl or biomass value, but not an abundance.  

Line 321 It is not normal pH range. It shifted to alkali range.

Line 394-395 Microcystins in the supernatant (90% methanol) can not be retained on the SPE cartridge material C18, they will be eluted with such a solvent.

Line 423 Could you mention the type of the used mass-spectrometer and its model.

Line 424 What was a purpose of using a wide range (m/z 500-1150) in full scan regime, if you are going to detect only one structure variant of microcystins (using SRM mode)? You could detect other structural microcystin variants using full scan regime. What is about the other microcystin structures?

Line 436 The recovery of the extraction should be determined using the same range of concentration which is expected in the samples. 100 mkg/L used for the spike experiment is a very high concentration for the environment.

In total, the discussion part is consisted of 50% results. It should be rewritten.

It could be better to discuss two groups of water bodies (component 1 and component 2) which was divided in the beginning of the Discussion.

I suggest to add comparison of your results with results of other authors on water bodies of the same latitude and similar environmental conditions?

Reviewer 2 Report

This is the first information on cyanobacteria and cyanotoxin production , especially microcystin-LR, in Cuban surface waters and therefore can be interested  to the readers and researchers  dealing with the global problem of cyanobacterial blooms. However, the manuscript needs a lot of improvements and corrections.  There are only some of remarks  indicated below:

1/English must be corrected and style of writing, too. There are  many sentences unclear. Writing of many terms should be unified in the text;  examples: microcystin- LR, names of species  in italic font, physico-chemical, etc. Abbreviation for microcystin should be rather MC-LR as  more frequently used in literature instead of MYC-LR.

2/Introduction:  other factors that infleunce development of cyanobacteria such as agricultural  activities and concentration of biogenic substances should be  indicated .

3/ Material and Methods: description of study area should contain information on the surface area and depth of reservoirs.  The map with reservoirs' location should be moved to this section. Table S1 is not needed, only description of the catchments etc can be included. The section between lines 66 and 90 should be removed from Results.

There is lack of information how frequently sampling was carried out and exactly in which terms. The species composion of cyanobacteria is strongly connected with the seasons.  Many cyanobacteria are colonies consisting of very small cells ( eg. Aphanizomenon, Microcystis) . How were the cells  counted ?

Results: In Figs 2,3 there is lack of information on the error bars nad determination repetitions.  Are the data expressed as mean values?  Titles of Figs 2-6 are not very informative.

Nutrients belong to chemical parameters of water , so it is not necessary to present them in a separate section. Fig 6 is completely unclear and Fig 7 is badly marked . What is PC? Instead of Fig 5 it would be more important to show a chromatogram with identified MC-LR.. Suplemmented Table S2  shows reachness of cyanobacterial species  not a diversity. The cyanobacteria indicated with (T) are rather potentially toxigenic , but it was not proved that toxic. There are many cyanobacterial populations which don't produce toxins. There was not performed any correlation between the abundance of toxigenic cyanobacteria and the production of MC-LR. What was the criterium  choosen to the marked with asterisk some species to describe them as  bloom-forming? Was it just theoretical of determined in the reservoirs? Generally,  results need re-elaboration. The quality of figures and tables is rather low.

Discussion is too long and should be focused more on the  main subject of the study (cyanobacteria and toxins). Some comments ato too trivial, eg. positive correlation between air temperature and surface water temp. It is obvious especially in shallow water bodies, but there is no information on the depth and type of water mixing in the reservoirs, etc.